# Feasibility and Preliminary Efficacy of American Elderberry Juice for Improving Cognition and Inflammation in Patients with Mild Cognitive Impairment

**DOI:** 10.3390/ijms25084352

**Published:** 2024-04-15

**Authors:** Ashley F. Curtis, Madison Musich, Amy N. Costa, Joshua Gonzales, Hyeri Gonzales, Bradley J. Ferguson, Briana Kille, Andrew L. Thomas, Xing Wei, Pei Liu, C. Michael Greenlief, Joel I. Shenker, David Q. Beversdorf

**Affiliations:** 1College of Nursing, University of South Florida, Tampa, FL 33620, USA; ashleycurtis@usf.edu (A.F.C.); amycosta@usf.edu (A.N.C.); 2Department of Psychological Sciences, University of Missouri, Columbia, MO 65201, USA; mmusich@mail.missouri.edu (M.M.); bkille01@gmail.com (B.K.); 3Department of Psychology, University of South Florida, Tampa, FL 33620, USA; 4School of Osteopathic Medicine, A. T. Still University, Kirksville, MO 63501, USA; sa198547@atsu.edu; 5Department of Internal Medicine, School of Medicine, Indiana University, Indianapolis, IN 46202, USA; 6School of Medicine, University of Missouri, Columbia, MO 65211, USA; hgx68@health.missouri.edu; 7Department of Neurology, University of Missouri, Columbia, MO 65211, USA; fergusonbj@health.missouri.edu (B.J.F.); shenkerj@health.missouri.edu (J.I.S.); 8Children’s Hospital Colorado, Aurora, CO 80045, USA; 9Division of Plant Science and Technology, University of Missouri, Southwest Research Extension and Education Center, Mt. Vernon, MO 65201, USA; thomasal@missouri.edu; 10Charles W. Gehrke Proteomics Center, Department of Chemistry, University of Missouri, Columbia, MO 65201, USA; xw4xf@mail.missouri.edu (X.W.); peiliu0824@gmail.com (P.L.); greenliefm@missouri.edu (C.M.G.); 11Department of Radiology, University of Missouri, Columbia, MO 65211, USA

**Keywords:** mild cognitive impairment, inflammatory, cognition, proteomics, elderberry, sambucus

## Abstract

Despite data showing that nutritional interventions high in antioxidant/anti-inflammatory properties (anthocyanin-rich foods, such as blueberries/elderberries) may decrease risk of memory loss and cognitive decline, evidence for such effects in mild cognitive impairment (MCI) is limited. This study examined preliminary effects of American elderberry (Sambucus nigra subsp. canadensis) juice on cognition and inflammatory markers in patients with MCI. In a randomized, double-blind, placebo–controlled trial, patients with MCI (*n* = 24, *M*_age_ = 76.33 ± 6.95) received American elderberry (*n* = 11) or placebo (*n* = 13) juice (5 mL orally 3 times a day) for 6 months. At baseline, 3 months, and 6 months, patients completed tasks measuring global cognition, verbal memory, language, visuospatial cognitive flexibility/problem solving, and memory. A subsample (*n* = 12, 7 elderberry/5 placebo) provided blood samples to measure serum inflammatory markers. Multilevel models examined effects of the condition (elderberry/placebo), time (baseline/3 months/6 months), and condition by time interactions on cognition/inflammation outcomes. Attrition rates for elderberry (18%) and placebo (15%) conditions were fairly low. The dosage compliance (elderberry—97%; placebo—97%) and completion of cognitive (elderberry—88%; placebo—87%) and blood-based (elderberry—100%; placebo—100%) assessments was high. Elderberry (not placebo) trended (*p* = 0.09) towards faster visuospatial problem solving performance from baseline to 6 months. For the elderberry condition, there were significant or significantly trending decreases over time across several markers of low-grade peripheral inflammation, including vasorin, prenylcysteine oxidase 1, and complement Factor D. Only one inflammatory marker showed an increase over time (alpha-2-macroglobin). In contrast, for the placebo, several inflammatory marker levels increased across time (L-lactate dehydrogenase B chain, complement Factor D), with one showing deceased levels over time (L-lactate dehydrogenase A chain). Daily elderberry juice consumption in patients with MCI is feasible and well tolerated and may provide some benefit to visuospatial cognitive flexibility. Preliminary findings suggest elderberry juice may reduce low-grade inflammation compared to a placebo–control. These promising findings support the need for larger, more definitive prospective studies with longer follow-ups to better understand mechanisms of action and the clinical utility of elderberries for potentially mitigating cognitive decline.

## 1. Introduction

Mild cognitive impairment (MCI) is considered the prodromal phase of dementia, with patients showing impairment in one or more cognitive domains, while activities of daily living remain preserved [1,2]. Approximately 10–15% of individuals with MCI develop dementia each year [3]. Therefore, evaluating interventions that can delay the progression of more severe cognitive decline and development of Alzheimer’s disease is of critical importance. Cognitive decline has been linked with inflammation [4,5], which has been proposed as an underlying mechanism contributing to Alzheimer’s disease and is a promising target for intervention [6]. Elderberries have anti-inflammatory and antioxidant properties [7,8], and show promising potential effects for improving cognition [9,10] and other areas of mental health [11,12,13] and related functioning [11,14]. However, to date, there have been no randomized controlled studies comparing the effects of elderberry consumption relative to a placebo–control in individuals with MCI. Evaluating the effect of a non-invasive nutritional elderberry intervention on cognition and inflammatory markers will help determine if and potentially how the course of the conversion of MCI to dementia can be mitigated.

Elderberry is one of the richest sources of polyphenols (including anthocyanins—red, purple, and blue-colored pigments [15]) and vitamins, compared to other berries with similar chemical compositions [16]. Prior studies have found that polyphenols, including anthocyanins, have a significant amount of antioxidant properties [17,18] and have been shown to reduce oxidative stress and inflammation [19]. In particular, anthocyanins have been shown to protect cells against oxidative damage [20] and free-radical-induced (damaging lipids and proteins) injury and diseases [18]. In particular, rodent studies have demonstrated the ability of anthocyanins to reduce oxidative-stress-based injury and disease because of their ability to neutralize free radicals [21]. Specifically, anthocyanins have been found to reduce mitochondrial oxidative stress, a major factor underlying pathology of many neurodegenerative diseases [22]. In further support of this, another rodent study found that a lingonberry extract, a berry that also has anthocyanins, improved cellular viability and reduced several markers of inflammation and oxidative stress, including cytokines [23]. Thus, the anthocyanin-rich food properties found in elderberries might be a viable option to protect against neurodegeneration in humans.

Neurodegenerative diseases, such as Alzheimer’s disease, are characterized by cellular death or apoptosis within specific regions of the brain [24]. It has been shown that anthocyanins are capable of crossing the blood–brain barrier, suggesting that anthocyanins may mitigate some of the damaging effects of neurodegeneration [24]. In vitro and in vivo chronic oxidative stress in the brain may deplete antioxidant capacity, and lead to the onset and progression of Alzheimer’s disease [25,26,27]. Further, animal models suggest that diets high in polyphenolic compounds prevent and reverse oxidative neurologic stress. For instance, a study in mice found that a diet rich in anthocyanins lessened neurodegeneration and improved memory compared to a control [28]. Another rodent study found that anthocyanin treatment protected against the worsening of memory [23]. In humans, one study found that the frequent consumption of fruit and vegetable juices, which were high in polyphenols, was associated with a substantially decreased risk for Alzheimer’s disease [29]. However, the specific type of juice that was consumed that reduced the risk of Alzheimer’s disease is unclear. In other work with middle-aged and older cognitively healthy adults, the consumption of a mixed berry juice (blueberries, blackcurrant, elderberry, lingonberries, strawberry, and tomatoes) daily for five weeks led to greater improvement in working memory relative to a control group [30]. Other studies have found improvement in memory in cognitively healthy older adults after the daily consumption of anthocyanin-rich blueberry juice [31] and concord grape juice [32]. These benefits may be due to the antioxidant and anti-inflammatory effects of anthocyanins that are found in berries [17,18,19].

Thus far, few studies have examined the effects of anthocyanins in a cognitively impaired population. One randomized placebo–controlled trial in older adults with MCI found that relative to a placebo juice, a polyphenol-rich grape and blueberry extract intervention led to greater 6-month improvements in speed of processing, visuospatial learning, and self-reported executive function [33]. Similarly, another study in older adults with self-reported cognitive decline found that a wild blueberry juice improved speed of processing to a greater degree than a placebo–control at 6 months [34]. However, the impact of a nutritional supplement, such as elderberry juice, on a range of cognitive domains, as well as inflammatory markers, has not been adequately explored in MCI. Thus, it remains unclear whether there is a clear causal relationship between foods or supplements rich in antioxidants and a decrease in relative risk for the development of more severe cognitive impairment (e.g., Alzheimer’s disease) [35]. Additionally, the mechanisms underlying this pathway (e.g., changes in circulating levels of specific inflammatory cytokines and/or preservation of behavioral cognitive abilities) remain to be fully determined. 

The present study aimed to examine the feasibility and preliminary effects of an American elderberry (Sambucus nigra subsp. canadensis) juice intervention (3x daily for 6 months) on cognitive performance (across a range of domains) and blood-based inflammatory markers in patients with MCI in a randomized, double-blind, placebo–controlled study. First, we hypothesized that in patients with MCI, attrition rates would be low and adherence to the elderberry juice intervention dosage, as well as cognitive and blood-based assessments at baseline, 3 months, and 6 months, would be feasible, as evidenced by high (>75%) dosage compliance and assessment completion rates. Second, we hypothesized that compared to a placebo–control condition, those receiving elderberry treatment would experience better cognitive function over time (baseline to 6 months). Finally, we hypothesized that patients with MCI receiving the elderberry intervention would show a greater reduction in inflammatory levels over time compared to the placebo–control condition.

## 2. Results

### 2.1. Participant Characteristics

Table 1 provides participant demographics, and Table 2 and Table 3 provide the main outcome values across time for each condition. Differences between elderberry and placebo–control conditions at baseline were evaluated by independent *t*-tests and chi-square tests at alpha-level 0.05 for continuous and categorical variables, respectively. There were no group differences observed for demographic and cognition variables at baseline (see Table 1 and Table 2). For proteomics outcomes, those receiving elderberry juice had significantly lower LDHA (*p* = 0.0495), A2M (*p* = 0.046), and PEDF (*p* = 0.02) levels at baseline, as shown in Table 3.

For CDR scores, all participants [*n* = 11 (100%) who were given elderberry juice; *n* = 13 (100%) who were given placebo–control juice] remained stable at 0.5 across timepoints. For CGI-C scores at 3 months, in those receiving elderberry, seven (64%) had scores indicating no change, two (18%) had scores indicating minimal improvement, and two (18%) did not have scores completed. At 6 months, two (18%) had scores indicating minimal worsening, five (45%) had scores indicating no change, one (9%) was recorded as having minimal improvement, and three (27%) were not completed. For the placebo–control and CGI-C scores at 3 months, two (15%) were recorded as minimally worse, seven (54%) had scores indicating no change, one (8%) showed minimal improvement, and three (23%) were not completed. At 6 months, three (23%) were recorded as minimally worse, seven (54%) were recorded as no change, one (8%) showed minimal improvement, and two (15%) were not completed. 

### 2.2. Feasibility

The CONSORT (Consolidated Standards of Reporting Trials) diagram is outlined in Figure 1. Of the 24 participants who were randomized [elderberry (*n* =11); placebo–control (*n* = 13)], 2 in the elderberry condition and 2 in the control condition did not complete 3 month follow-ups [due to gastrointestinal issues (*n* = 1), other health issues (*n* = 1), potential allergic reaction (*n* = 1), and no longer interested (*n* = 1); see CONSORT Figure 1]. All remaining participants completed 3-month (elderberry: 3 men, 6 women; placebo–control: 6 men, 5 women) and 6-month follow-ups. Thus, attrition rates were fairly low for elderberry (18% attrition) and placebo (15% attrition) conditions. 

Compliance with the intervention (elderberry) and control (placebo juice) dosages was high (see Table 1), with no significant difference observed between groups (*p* = 0.46). Cognitive assessment completion across the three timepoints was also high for the elderberry group (average of 88% of assessments completed, with a range of 33–100% completion across participants) and placebo–control (average of 87% of assessments completed, with a range of 33–100% completion across participants). 

For blood-based proteomic outcomes, out of the 12 participants (6 men, 6 women) in the sub-sample who were asked to complete blood draws across the three timepoints, completion was very high in the elderberry condition [out of 7 participants (3 men, 4 women) in this condition, an average of 100% of blood draws were completed, with a range of 100–100% across participants] and placebo–control [out of 5 participants (3 men, 2 women) in this condition, an average of 100% of blood draws were completed, with a range of 100–100% completion across participants].

### 2.3. Preliminary Efficacy

Table 2 provides the descriptive values for cognition outcome variables for each condition (elderberry vs. placebo–control) across timepoints (baseline, 3 months, 6 months) and the MLM model statistics. 

### 2.4. Cognitive Outcomes

Interactions. There was a trending interaction between the time and condition in the VPS—Mean Latency Correct (see Table 2). As shown in Figure 2, post hoc pairwise comparisons show that in the elderberry condition, solution latencies decreased between baseline and 6-month follow-up [*t* (39.6) = 1.73, *p* = 0.09, *g_av_* = 0.70, moderate effect], whereas there was no significant differences or trending towards significant differences between baseline and 3 months (*p* = 0.39) or 3 months and 6 months (*p* = 0.39). For the placebo–control condition, there were no changes in scores between baseline and 3 months (*p* = 0.82), 3 months and 6 months (*p* = 0.64), or baseline and 6 months (*p* = 0.44).

Main effects. As expected for a progressive memory impairment, there were trending main effects of time on HVLT—Recognition # Hits (see Table 2), with scores decreasing between baseline and 6 months [*t* (33.60) = 2.06, *p* = 0.047, *g_av_* = 0.42, small effect] and 3 months and 6 months [*t* (31.30) = 2.38, *p* = 0.02, *g_av_* = 0.49, small effect], while there was no difference observed between baseline and 3 months (*p* = 0.69). Similarly, there was a main effect of time for HVLT—Discrimination Index (see Table 2) with scores decreasing between baseline and 6 months [*t* (33.80) = 1.87, *p* = 0.07, *g_av_* = 0.44, small effect], whereas there was no difference observed between baseline and 3 months (*p* = 0.46) and between 3 and 6 months (*p* = 0.26). There was a significant main effect of time for BNT (see Table 2), with scores generally increasing between baseline and 3 months [*t* (37.00) = −2.71, *p* = 0.01, *g_av_* = 0.34, small effect], and baseline and 6 months [*t* (37.00) = -2.93, *p* = 0.006, *g_av_* = 0.34, small effect], while there was no difference observed between 3 and 6 months (*p* = 0.83). 

### 2.5. Blood-Based Inflammatory Markers

Table 3 provides descriptive values for blood-based inflammatory outcomes for each condition (elderberry vs. placebo–control) across timepoints (baseline, 3 months, 6 months) and the MLM model statistics. 

Interactions. There were significant interactions between the time and condition for LDHA and Factor D (see Table 3). There were also trends towards significant interactions between the time and condition for LDHB, A2M, vasorin, PEDF, and PCYOX1 (see Table 3). 

As shown in Figure 3a, for LDHA, post hoc pairwise comparisons revealed that levels were significantly higher in the placebo–control condition than the elderberry condition at baseline [*t* (127.00) = 2.18, *p* = 0.03, *g_av_* = 1.20, large effect]; however, there were no longer significant differences between conditions at 3 months (*p* = 0.57) or 6 months (*p* = 0.95). Regarding changes within conditions, in the placebo–control, post hoc comparisons showed that levels were lower than baseline at 3 months [*t* (20.00) = 2.53, *p* = 0.02, *g_av_* = 0.91, large effect] and 6 months [*t* (20.00) = 3.30, *p* = 0.004, *g_av_* = 1.22, large effect], whereas levels were maintained between 3 and 6 months (*p* = 0.45). There were no observed changes over time for LDHA levels in the elderberry condition in post hoc comparisons (*ps* ranged from 0.68 to 0.91). 

As shown in Figure 3b, for LDHB, post hoc comparisons showed that placebo–control levels at 6 months were significantly lower than baseline [*t* (20.00) = 2.81, *p* = 0.01, *g_av_* = 1.45, large effect], and trended towards significance for being lower than 3 months [*t* (20.00) = 2.03, *p* = 0.06, *g_av_* = 0.33, small effect]. At 6 months, results showed a trend towards significance for lower values in the placebo–control group relative to elderberry [*t* (32.00) = −1.84, *p* = 0.08, *g_av_* = 0.95, large effect]. 

As shown in Figure 3c, A2M levels for the elderberry condition at 6 months were significantly higher than baseline [*t* (20.00) = −2.73, *p* = 0.01, *g_av_* = 0.78, moderate–large effect], and trending towards significance for higher values than 3 months [*t* (20.00) = −1.99, *p* = 0.06, *g_av_* = 0.24, small effect] in post hoc comparisons. No change in levels across time was observed for the placebo–control condition (*ps* ranged from 0.22 to 0.85), and conditions did not significantly differ from each other across timepoints (*ps* ranged from 0.13 to 0.63) in post hoc comparisons. 

As shown in Figure 3d, vasorin levels in post hoc comparisons for the elderberry condition were significantly lower than baseline at 3 months [*t* (20.00) = 2.59, *p* = 0.02, *g_av_* = 0.87, large effect] and 6 months [*t* (20.00) = 2.86, *p* = 0.01, *g_av_* = 0.72, moderate effect]. In the placebo–control condition, post hoc comparisons showed that vasorin levels did not significantly differ over time (*ps* ranged from 0.55 to 0.79). Levels between conditions did not significantly differ across timepoints (*ps* ranged from 0.11 to 0.81). 

As shown in Figure 3e, for PCYOX1, post hoc comparisons showed that levels were trending towards significantly higher levels in the elderberry condition compared to the placebo–control condition at baseline [*t* (100.90) = 1.77, *p* = 0.08, *g_av_* = 1.14, large effect], while there were no differences between conditions observed at 3 (*p* = 0.55) and 6 months (*p* = 0.83). Regarding within-condition comparisons, PCYOX1 level increases from baseline to 6 months were trending for the elderberry condition [*t* (20.00) = −1.94, *p* = 0.07, *g_av_* = 0.81, large effect]. No change across time was observed for the placebo–control condition (*ps* ranged from 0.56 to 0.81). 

As shown in Figure 3f, for Factor D, levels significantly increased from 3 months to 6 months for the placebo–control condition [*t* (20.00) = −2.15, *p* = 0.04, *g_av_* = 1.50, large effect], while for the elderberry condition, Factor D levels trended towards significance [*t* (20.00) = 2.08, *p* = 0.05, *g_av_* = 0.80, large effect] for a decrease from baseline to 6 months. Factor D levels were also significantly trending towards lower levels in the placebo–control group relative to the elderberry condition at 3 months [*t* (34.2) = −0.54, *p* = 0.097, *g_av_* = 0.33, small effect), while placebo–control levels were significantly higher than elderberry at 6 months [*t* (34.20) = 2.37, *p* = 0.02, *g_av_* = 1.15, large effect]. 

For PDEF levels, the only difference between conditions was at baseline, with a trend towards significance of higher levels in the placebo –control group than elderberry [*t* (100.90) = 1.77, *p* = 0.08, *g_av_* = 1.44, large effect]. Other pairwise comparisons between conditions and over time were non-significant (*ps* ranged from 0.16 to 0.54).

Main effects. There were significant main effects of time for LDHA, LDHB, C4-A, and C4-B (see Table 3). For LDHA, levels generally decreased from baseline to 3 months [*t* (20.00) = 2.19, *p* = 0.04, *g_av_* = 0.58, moderate effect] and from baseline to 6 months [*t* (20.00) = 2.72, *p* = 0.01, *g_av_* = 0.69, moderate effect], while levels did not significantly change between 3 months and 6 months (*p* = 0.61). LDHB levels generally decreased from baseline to 3 months [*t* (20.00) = 2.31, *p* = 0.03, *g_av_* = 0.23, small effect], while there was no significant change observed between baseline and 3 months (*p* = 0.51) or between 3 months and 6 months (*p* = 0.62). Conversely, C4-A levels generally increased from baseline to 3 months [*t* (20.00) = −3.48, *p* = 0.002, *g_av_* = 0.59, moderate effect] and 6 months [*t* (20.00) = −4.23, *p* < 0.001, *g_av_* = 0.74, large effect], while there was no significant change observed between 3 months and 6 months (*p* = 0.47). Similarly, C4-B levels generally increased from baseline to 3 months [*t* (20.00) = −3.41, *p* = 0.003, *g_av_* = 0.59, moderate effect] and 6 months [*t* (20.00) = −3.96, *p* < 0.001, *g_av_* = 0.70, moderate effect], while there was no significant change observed between 3 months and 6 months (*p* = 0.59).

## 3. Discussion

The present randomized, double-blinded, placebo–controlled trial evaluated the feasibility and preliminary efficacy of an American elderberry juice intervention relative to a placebo juice control on cognitive and blood-based markers of inflammation in patients with MCI. Findings revealed that the 3x daily dosage of elderberry was feasible and well tolerated at 6 months, as evidenced by low attrition, high dosage adherence, and high rates of assessment completion. There were only limited effects of the elderberry juice versus control on cognition, with a trending greater elderberry benefit observed for visuospatial cognitive flexibility (VPS mean latency scores) at 6 months compared to the placebo–control. Greater elderberry-related improvement across other domains (verbal memory, language, visuospatial construction, visual memory, verbal cognitive flexibility, global cognition) was not observed. Preliminary findings for the elderberry condition also showed significant or trending decreases over time across several markers of low-grade peripheral inflammation, including vasorin, PCYOX1, and Factor D, while only one inflammatory marker showed an increase over time (A2M). In contrast, for the placebo condition, several inflammatory marker levels increased across time (LDHB, Factor D), with one showing decreased levels over time (LDHA). 

As predicted, the 3x daily dosage of elderberry juice over 6 months and associated cognitive and blood-based inflammatory marker assessment were feasible in patients with MCI. Six-month attrition rates (18% for elderberry; 15% for placebo) were comparable to other randomized clinical trials examining blueberry extract interventions in cognitively impaired populations across similar 6-month timeframes but with fewer (2x) daily doses (blueberry attrition rate ranged from 12% [33] to 14% [34]; placebo attrition rate ranged from 14% [34] to 20%). Reasons for drop-out in the elderberry condition (minor gastrointestinal issue or co-morbid medical occurrence) are also comparable to previous drop-out reasons reported in prior blueberry extract studies in patients with MCI [34]. 

Our second hypothesis that elderberry juice consumption would lead to greater cognitive improvement than a placebo–control was only partially supported, as we only observed a trend towards greater elderberry improvement in one domain—visuospatial cognitive flexibility. This is in partial agreement with a previous study that found that a blueberry extract improves visuospatial learning in patients with MCI [33]. We offer several potential explanations for the present results. First, executive dysfunction related to visuospatial planning and problem solving encompasses the cumulative burden of frontal and parietal lobe damage [38,39,40]. It is possible that elderberry consumption selectively reduces mitochondrial oxidative stress [23] in these frontal/parietal regions and improves associated cognitive function. Prior preclinical work shows a selective benefit of anthocyanins on frontal and hippocampal free radical reduction [41]. However, it is unclear if the burden of neuropathological changes in participants in the present study was primarily in the medial temporal lobe, and thus contributing to memory deficits [39]. Additionally, prior work showed that relative to other cognitive tasks, visuospatial problem solving had better discriminative ability to detect non-memory impairments in MCI [42]. This raises the possibility that for patients in the present study, the cumulative oxidative neuropathological burden in frontal and parietal regions could have been more sensitive to anthocyanin-related free radical reduction [23]. Second, prior work also showed a unique benefit to cerebrovascular function in parietal regions after longer term (12 weeks) polyphenol (blueberry concentrate) consumption [43]. Therefore, it is also possible that elderberry consumption leads to better cerebrovascular parietal function and consequently more cognitive task-related (i.e., visuospatial problem solving, which involves both parietal and frontal regions [38]) neural activation. Finally, unlike previous findings, we did not observe improvement across memory or other aspects of executive function [33]. Other studies have also found improvement in speed of processing as a result of grape/blueberry extract consumption [33,34]. Although we did not measure speed of processing/reaction time per se in this study, it is notable that the only metric in our visuospatial problem solving task where we saw improvement was for the speed in which visuospatial cognitive flexibility tasks were completed. Therefore, future work should examine whether elderberry juice also impacts speed of processing/reaction time tasks more generally.

Our third hypothesis that elderberry juice would reduce levels of blood-based inflammation to a greater degree than a placebo–control was partially supported. The overall pattern of results shows that relative to a placebo–control, elderberry juice led to more consistently favorable reductions in inflammation marker levels. Specifically, results point towards vasorin, PCYOXI, and Factor D as potential inflammatory targets and/or elderberry-related mechanisms of action. Vasorin plays a role in arterial response to injury [44] and PCYOX1 is a pro-oxidant enzyme known to be involved in heightened oxidative stress, tissue inflammation, and thickening/hardening of the arteries of atherosclerosis [45,46]. Emerging work has also suggested that PCYOX1 gene expression may play a key role in neurodegenerative pathology related to synaptic dysfunction [47]. Factor D is associated with low-grade inflammation and endothelial dysfunction related to artery contraction and relaxation [48]. There is a growing field of research suggesting a link between peripheral inflammatory markers and central inflammatory processes in Alzheimer’s disease [49,50]. Therefore, these preliminary results suggest that anthocyanin/elderberry-related impact on peripheral blood-based inflammation may also impact central inflammation, and/or be related to the cognitive benefits observed. Future work in larger MCI patient samples should test blood-based markers of neurodegeneration such as tau and Aβ (40/42) [51] and evaluate whether elderberry-related inflammatory changes are associated with cognitive change, in order to determine mechanisms of action. 

### 3.1. Clinical Implications

While limited as a feasibility study, there are several potential future clinical implications that could arise from the present findings. Executive dysfunction related to visuospatial planning and problem solving encompasses the cumulative burden of frontal and parietal lobe damage [38], which is known to be affected early in Alzheimer’s disease [42,52]. Thus, the trending benefit of elderberry juice on visuospatial problem solving/cognitive flexibility suggests that it may have clinical utility as a non-invasive nutritional supplement to help mitigate the progression of cognitive dysfunction profiles in prodromal Alzheimer’s disease. Further, given the more consistent pattern of the elderberry-juice-related reduction in peripheral inflammation in the present study, and other findings suggesting that elevated peripheral inflammation may be an important marker of elevated risk and onset of Alzheimer’s disease [49], the present findings also provide promising preliminary support for the potential clinical utility of elderberry juice for mitigating the progression of Alzheimer’s-related neuropathology. 

### 3.2. Limitations and Future Directions

There are several limitations of the present study. First, the sample size was small, particularly for the sub-set of participants who completed blood draws. Therefore, follow-up studies with larger sample sizes are needed. Second, it is possible that the timeframe of elderberry consumption (6 months) was not long enough to prompt significant effects on a broader range of cognitive outcomes. Given that several of the inflammatory markers show trending towards significance or significant decreases at 6 months, it is possible that cognitive improvement could be secondary to this change and manifest at later timepoints (e.g., 9 months, 1 year). It is important to note that we were not powered to examine whether associations between elderberry inflammatory marker change were linked to cognitive change (given we had less than 10 participants in each group that completed the blood-based assessment [53]). However, in future larger-scaled studies, it will be critical to examine if inflammation reduction may drive future cognitive improvement. Third, given other studies showing improvement in a grape/blueberry extract condition on reaction time [33,34] and self-reported executive function [33] in patients with MCI, future work may wish to examine whether elderberry juice may impact a broader range of objective and self-reported cognitive outcomes. The inclusion of self-reported cognitive outcomes may be particularly important, given that self-reported cognitive complaints have been associated with Alzheimer’s disease risk and pathology [54,55]. On a related note, given the known impact of nutritional interventions rich in anthocyanin/anti-inflammatory properties on other aspects of mental health (depression, anxiety, and stress [11,12,13]) and sleep health [11,14] in adults without cognitive impairment, future studies in MCI should also examine the impact of American elderberry juice on a broader range of outcomes beyond cognition and inflammation. Fourth, the present study did not assess the influence of other factors that may have anti-inflammatory properties and impact blood-based inflammatory markers and cognitive functioning in MCI, such as diets (e.g., Mediterranean diet rich in omega-3 fatty acids, antioxidants, and polyphenols [56]), medications, and vitamins [57,58]). Additionally, given known sex differences in trajectories of cognition [59,60] and low-grade inflammation [61] in older adults, future work should also examine the sex-specific impact of elderberry juice on cognitive and inflammation outcomes. Future studies could also examine interactions between elderberry juice and other factors to determine whether a combination of interventions can have a more potent effect [62,63]. Finally, given that 100% of the study participants were White/Caucasian, it is critical that future studies examine the effects of elderberry juice in more racially and ethnically diverse samples.

## 4. Materials and Methods

### 4.1. Participants

Participants were recruited via physician referral (D.Q.B., J.S.) from two Memory Disorder Clinics in Columbia, MO. Inclusion criteria were (1) aged 50+ years, (2) diagnosed with MCI, (3) a Clinical Dementia Rating Scale (CDR) score of 0.5 (administered by study physicians D.Q.B. or J.S.), (4) a Mini-Mental Status Examination (MMSE [33]) score of 24+/30, (5) reported no known sensitivity or allergy to elderberry products, and (6) no presence of any health condition that in the clinical experience of the investigators might impair their ability to complete this study. Participants were excluded if they met one or more of the following criteria: known allergy to honeysuckle (closely related to Sambucus), a current diagnosis of diabetes, the presence of a bleeding disorder, currently pregnant, currently undergoing prescribed changes to other medications that might affect cognitive performance, the presence of any comorbid medical condition that would impair the patient’s ability to complete study procedures (e.g., terminal illness, comorbid major psychiatric disorders such as schizophrenia, or substance use disorder), and the presence of other neurodegenerative diseases (e.g., Parkinson’s disease). All study procedures were approved by the University of Missouri Institutional Review Board. The clinical trial was pre-registered (NCT02414607; PI: Beversdorf).

### 4.2. Procedure and Study Design

The study design was a randomized, double-blinded, placebo–controlled trial. Participants were randomly assigned to one of two conditions: elderberry juice intervention or a placebo–control juice. Participants were instructed to take 5 mL of the juice by the mouth three times daily for 6 months. The elderberry juice was a commercially available 100% pure American elderberry product that was stabilized with citric acid (River Hills Harvest, Hartsburg, MO, USA; see Appendix A) and contained 15.9 mg of cyanidin–glucose equivalents (i.e., main anthocyanin in elderberry), resembling well-tolerated doses utilized in previous research that examined effects of elderberry juice on lipoproteins (see Appendix A [7]). The placebo–control juice contained flavored liquid with no nutritional content. All participants were provided sufficient juice at each laboratory visit to last until the next laboratory visit. Participants were instructed to complete a daily log documenting their juice intake. Juice containers were brought back to each laboratory visit to confirm self-reported juice consumption by the research assessor. Participants were excluded if they consumed less than 75% of their scheduled dose. Participants completed assessments of cognition and a sub-sample (see Results (Section 2)) provided blood samples for proteomic bio-inflammatory markers (see Measures (Section 4.3.3)) at their first appointment (i.e., baseline). They were then retested on these cognitive measures and provided additional blood samples at 3- and 6-month appointments. 

### 4.3. Measures

#### 4.3.1. Clinical Assessments

Clinical Dementia Rating (CDR). The CDR is a clinical scale used to evaluate six domains of cognitive and functional performance (memory, orientation, judgment and problem solving, community affairs, home and hobbies, personal care) that evaluate the progression of dementia [64,65]. Impairments are assessed through a semi-structured interview with the patient and informant, on a five-point scale (0 = no impairment, 0.5 = questionable impairment, 1 = mild impairment, 2 = moderate impairment, 3 = severe impairment). The global CDR score represents a 5-point ordinal scale, where a global CDR score of 0.5 is indicative of MCI, characterized by significant memory disturbances but intact ability to perform everyday tasks [66]. 

Clinical Global Impression of Change-Clinician (CGI-C). The CGI-C [67] asks clinicians to rate the extent to which a patient has changed from 1 (markedly worse) to 7 (markedly improved) across several areas of functioning (overall medical problems, thinking, feelings and behavior, daily activities, involvement in social or community activities, and overall functioning) as compared to the baseline visit. The CGI-C is commonly used in clinical trial research on behavioral and pharmacological interventions for dementia and is a more sensitive measure for evaluating change compared to other assessments [67]. Total CGI-C scores were provided by the clinician at 3 months and 6 months. 

#### 4.3.2. Cognitive Assessments

Mini-Mental State Exam (MMSE). The MMSE is a brief assessment of cognitive status [68]. The MMSE assesses seven cognitive functions: orientation to time, orientation to place, three-word registration, attention and calculation, three-word recall, language, and visual construction. The total score ranges from 0 to 30, with lower scores indicating worse global cognitive function. A cut-off score of <24 indicates moderate to severe cognitive impairment [69]. 

Hopkins Verbal Learning Test (HVLT). The HVLT measures verbal learning and memory [70]. The first part of this assessment relates to free recall. A 12-item word list with 4 words from each of the 3 semantic categories (i.e., ‘precious stones’, ‘human shelter’, ‘animals’) was orally presented and participants were instructed to recall, in any order, the list of words. This list of words was repeated three times, and an HVLT—Free Recall average score across trials 1–3 was calculated (higher scores indicating better immediate verbal memory). After a 20 min delay, participants were asked to recall the word list again and an HVLT—Delayed Free Recall score was computed (higher scores representing better long-term verbal memory). The second part of this assessment was related to recognition memory. In the recognition phase, a list of 24 words were orally presented, consisting of the 12 original words (true positive), 6 distractors from the same semantic categories (false-positive-related), and 6 unrelated distractors (false-positive-unrelated). Participants indicated whether the word was a part of the original list of words or not (yes/no). The following outcomes were computed: HVLT—Recognition Number (#) Hits (total number of correct true positive words recognized; higher scores indicating better performance), HVLT—Recognition False Alarm (FA)-Related (total false-positive-related words recognized; lower scores indicating better performance), and HVLT—Recognition FA-Unrelated (total number of false-positive-unrelated words recognized; lower scores indicating better performance). HVLT—Discrimination Index was also calculated by subtracting HVLT—Recognition # Hits from the total number of HVLT—Recognition FA (combined across related and unrelated), with higher scores indicating better performance. Alternate forms of the HVLT were used for each assessment timepoint.

Boston Naming Test (BNT). The BNT was used as a measure of language [71]. Participants are shown 60 line-drawings of objects with increasing difficulty, ranging from simple, high-frequency vocabulary words (e.g., comb) to rare words (e.g., abacus). Participants are asked to provide the common name for the line-drawing within 20 s. If participants are not able to provide the answer, two types of cues are provided: a semantic cue (e.g., the semantic cue for a helicopter is “used for air travel”), or a phonetic cue (e.g., “moo” for moose). The number of correct answers was computed for those given without a cue, with a semantic cue, and with a phonetic cue. The total correct answers were calculated, with higher scores indicating better performance. Alternate forms of the BNT were used for each assessment timepoint. 

Rey–Osterrieth Complex Figure Test (Rey CFT). The Rey CFT is used to evaluate visuospatial constructional ability and visual memory [72]. In the copy trial, participants are asked to copy a complex image on a separate sheet of paper as accurately as possible. Correctly copied elements receive points, with a maximum score of 36 points, and higher scores in this first copy trial (REY CFT—Copy Total) represent better visuospatial constructional ability. In the delayed recall trial, participants are asked to redraw the complex image from memory after a 30 min period, with correctly copied elements receiving up to 36 points, and higher scores (REY CFT—Delayed Copy Total) represent better visuospatial memory. Alternate versions of Rey CFT were used for each timepoint.

Anagrams. Anagram problem solving tasks were used as an assessment of verbal cognitive flexibility and convergent creativity [73,74]. Participants received one of two sets of 20 anagrams and were asked to unscramble each in a maximum of 120 s (e.g., the solution for “RPPEA” is “PAPER”). The anagrams that were administered consisted of 14 mildly challenging (5-letter anagrams) and 6 higher difficulty (7-letter anagrams) items. The total number of correct items was computed (Anagrams—Total Correct), with higher scores representing better verbal cognitive flexibility. Latency to the solution (Anagrams—Latency Total) was also measured (in seconds), with failed solutions recorded as 120 s. Lower Anagrams—Latency Total scores represent better verbal cognitive flexibility. Alternate forms of anagrams were used for each timepoint.

Visuospatial Problem Solving (VPS). The VPS task is a visuospatial task that involves cognitive flexibility and convergent creativity, adapted from previous work using the matchstick problems [38,75]. Participants were presented with lines/matchsticks and were instructed to remove or mentally move a line/matchstick to construct a specific configuration to solve the presented problem. A total of six problems were presented (e.g., “move 3 matches to make 5 small squares; every match must be part of some square”). The total number of correct problems solved was computed (VPS—Total Correct), with higher scores indicating better performance. Mean latency (in seconds) of solving correct problems was also computed (VPS—Mean Latency Correct), with lower values representing better performance. Alternate forms were also used at each timepoint.

#### 4.3.3. Blood-Based Biomarkers of Inflammation (Proteomic Outcomes)

Participants underwent a blood draw (5 mL) at the Clinical Research Center at the University of Missouri. Participants were instructed to fast (no eating or drinking anything other than water) for 8 h prior to the blood draw. Samples were transported to the Proteomics Center at the University of Missouri for additional processing. Blood samples were separated by centrifugation at 4 °C for 15 min at 2000 g in a Jouan Refrigerated Centrifuge (CR-412, Thermo Fisher Scientific, Pittsburgh, PA, USA) to obtain serum. 

Serum samples for proteomic analyses were subjected to albumin and IgG depletion using ProteoPrep^®^ Immunoaffinity Albumin and IgG Depletion Kit (Sigma-Aldrich, St. Louis, MO, USA) to eliminate these abundant proteins. The resulting depleted solution was precipitated with acetone and resuspended in a solution containing 6 M urea, 2 M thiourea, and 100 mM ammonium bicarbonate. Subsequently, the proteins were quantified, and 200 μg proteins was reduced and alkylated before digested by trypsin. The digested peptides were purified by Pierce C18 tips, then were lyophilized, and were resuspended in 5/0.1% acetonitrile/formic acid. The suspended peptide was analyzed on a timsTOF pro mass spectrometer, which was connected to a Bruker nonElute system. The peptide (0.80 μg) was separated on a C18 column (15 cm × 75 μm, 1.9 μm, Bruker) with a step gradient including an initial 3%B (A: 0.1% formic acid in water, B: 99.9% acetonitrile, 0.1% formic acid), followed with a 65 min ramp to 30%B. Subsequently, a 30–50%B was conducted over 10 min, and a gradient of 50%B to 80%B for 7.5 min with a total run time of 90 min. The MS data were collected in the 100–1700 m/z range. Each TIMS cycle during the MS/MS data collection included 1 MS scan and the average 10 PASEF MS/MS scans.

The acquired data were submitted to the PEAKS10 search engine for protein identifications using the Uniprot (https://www.uniprot.org/ (accessed on 20 April 2020)) and NCBI (https://www.ncbi.nlm.nih.gov/protein (accessed on 20 April 2020)) homo sapiens databases. Data were searched using the following criteria: trypsin enzyme, allowing 1 missed cleavage, carbamidomethyl cysteine as fixed modification, oxidized methionine, and deamidation of asparagine and glutamine as a variable modification. Mass tolerances were set at 50 ppm for precursor ions and 0.5 Da for fragmented ions. Following the search, duplicated proteins were removed, and a little more than 4000 protein groups were identified. The proteins were further filtered by 2+ unique peptides and >20 significant proteins (1334 proteins). Common proteins between samples were identified and those with 90+% overlap were retained and their relative concentrations were quantified (734 proteins). Proteins were then analyzed using multilevel modeling (MLM) to determine statistically significant proteins for the analysis (see Data Analysis for further statistical analysis methodology). The proteins whose relative concentrations significantly changed included L-lactate dehydrogenase A (LDHA) and B (LDHB) chains, alpha-2-macroglobulin (A2M), vasorin, pigment epithelium-derived factor (PEDF), complement C4-A (C4-A) and C4-B (C4-B), prenylcysteine oxidase 1 (PCYOXY1), and complement Factor D (Factor D).

### 4.4. Data Analysis

#### 4.4.1. Baseline Demographics and Clinical Characteristics

Differences between elderberry and placebo conditions were evaluated using independent sample *t*-tests for continuous variables and chi-square analyses for categorical variables. 

For categorical CDR and CGI-C clinical assessments, the frequency and percentage of participants with each score option were computed for elderberry and placebo–control conditions across each timepoint (baseline, 3 months, and 6 months for CDR; 3 months and 6 months for CGI-C).

#### 4.4.2. Feasibility

Attrition rates, intervention condition dosage adherence, and percent completion of study assessments were used to evaluate feasibility of the study protocol.

#### 4.4.3. Preliminary Efficacy

For each cognition and proteomic biomarker of the inflammation outcome variable, a priori planned analyses were conducted in R [76] using MLM (via lmer) [77] to test fixed effects of the condition (elderberry versus placebo–control; coded with placebo–control as reference level), time (baseline, 3-month follow-up, 6-month follow-up), and condition by time interaction with participants nested within the condition. Given the present study’s small and uneven sample sizes across conditions for repeated measures data, MLM (i.e., linear mixed effect modeling) with restricted maximum likelihood estimation [78,79] was used to reduce Type I error inflation. MLM also accommodates and adjusts for missing data, including all level 1 (within-group variables) and level 2 (between-group variables) data, in analyses to retain the sample size, while mixed analysis of variance (ANOVA) statistical models use list-wise deletion to reduce the sample size [79]. Given the preliminary nature of this study, both significant (*p* < 0.05) and trending (*p* < 0.10) [80] main effects and interactions were further clarified by post hoc least square means pairwise comparisons via dfflsmeans [81] to examine individual associations of the group and/or time with each significant cognition and inflammation outcome variable. Kenward–Roger correction post hoc pairwise was used to protect against an inflated Type I error rate in a small sample size [79]. Due to the preliminary nature of this study, family-wise error corrections were not implemented in our analyses and we accepted the false positive risk [82]. For MLM interaction terms, *η*^2^ was used to describe effect sizes, with qualifications as follows: small (*η*^2^ = 0.01), moderate (*η*^2^ = 0.06), and large (*η*^2^ = 0.14) [36,37]. For pairwise comparisons, average Hedges’ g (*g_av_*) was used to calculate effect sizes, with qualifications as follows: small (*g_av_* = 0.20), moderate (*g_av_* = 0.50), and large (*g_av_* = 0.80) [36].

## 5. Conclusions

Preliminary findings show that daily elderberry juice consumption for 6 months in patients with MCI is feasible and well tolerated and may provide some benefit to visuospatial cognitive flexibility. Patterns of preliminary blood-based inflammatory markers suggest that elderberry juice may lead to the reduction in low-grade inflammation compared with a placebo–control. These promising preliminary findings provide support for larger, more definitive prospective studies with longer follow-ups to better understand mechanisms of action and the clinical utility of elderberries for potentially mitigating cognitive decline.

## Figures and Tables

**Figure 1 ijms-25-04352-f001:**
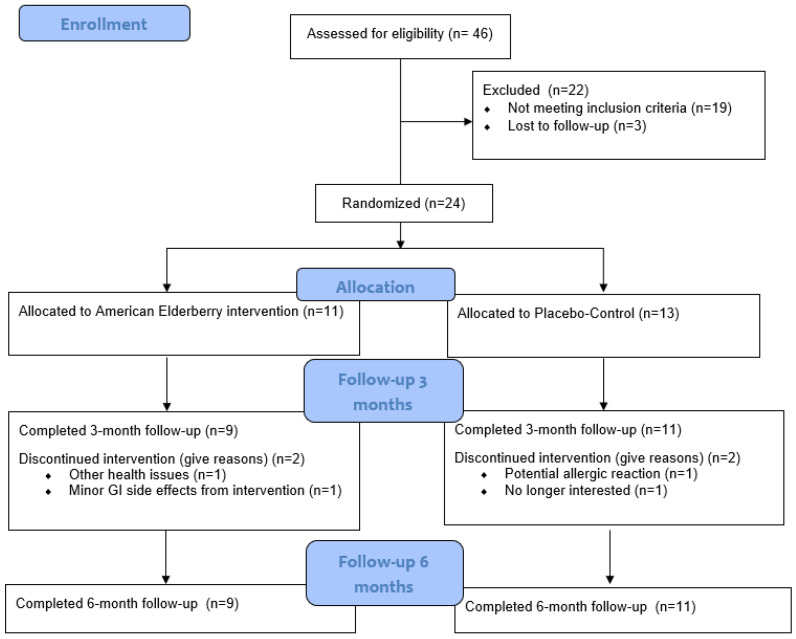
Participant recruitment CONSORT diagram.

**Figure 2 ijms-25-04352-f002:**
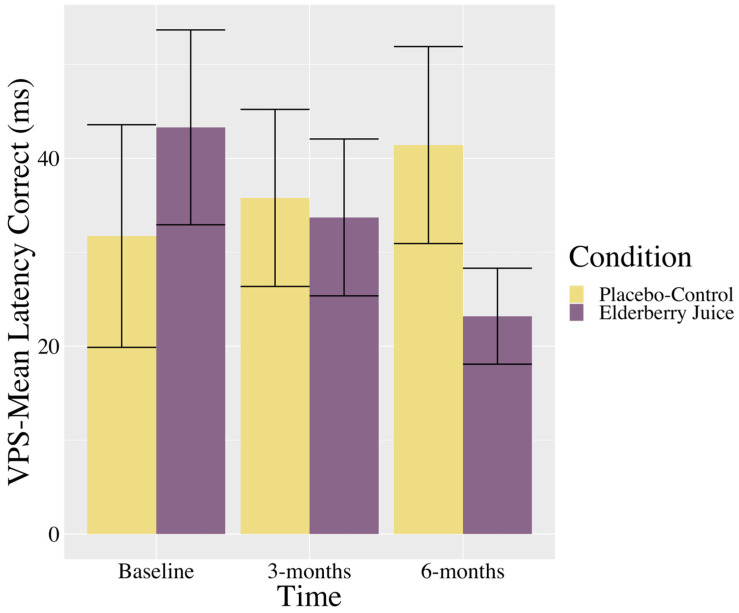
Changes in visuospatial problem solving (mean latency in correct trials) across time for American elderberry versus placebo–control in patients with MCI. *Note*. VPS = Visuospatial Problem Solving; error bars = SE.

**Figure 3 ijms-25-04352-f003:**
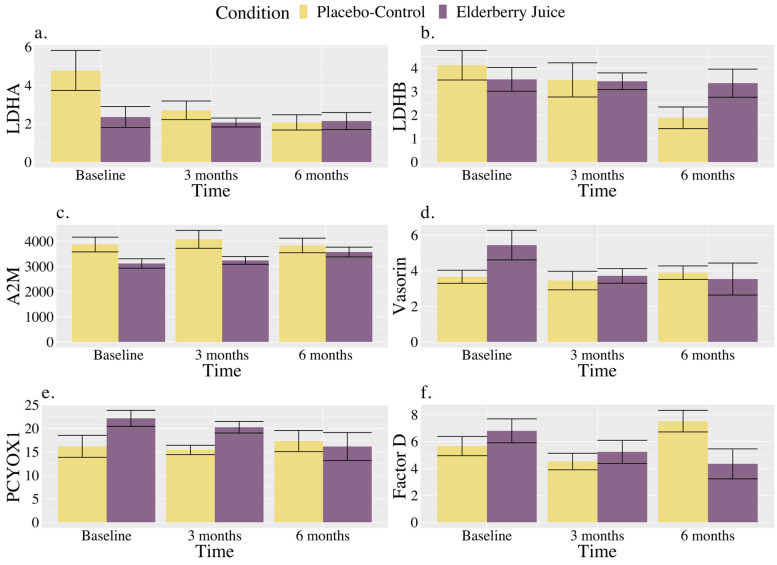
Changes in blood-based markers of inflammation across time for American elderberry versus placebo–control in patients with MCI for (**a**) LDHA, (**b**) LDHB, (**c**) A2M, (**d**) Vasorin, (**e**) PCYOX1, and (**f**) Factor D. *Note*. Blood-based inflammatory proteomic biomarkers were measured as a relative concentration. LDHA = L-lactate dehydrogenase A chain; LDHB = L-lactate dehydrogenase B chain; A2M = alpha-2-macroglobulin; PCYOXY1 = prenylcysteine oxidase 1; Factor D = complement Factor D; error bar = standard error.

**Table 1 ijms-25-04352-t001:** Demographics at Baseline in Participants with Mild Cognitive Impairment Assigned to American Elderberry Intervention or Placebo–Control (*n* = 24).

	Elderberry (*n* = 11)	Placebo–Control (*n* = 13)	Group Comparisons at Baseline
*Variable*	*M (SD)*	*M (SD)*	*p*
Age	76.45 (8.47)	76.23 (7.73)	0.94
Sex (M:F)	3:8	7:6	0.37
Race/Ethnicity (*n*, %)			0.99
White/Caucasian	(11, 100%)	(13, 100%)	--
Black/African American	(0, 0%)	(0, 0%)	--
Native American/ American Indian	(0, 0%)	(0, 0%)	--
Education (*n*, %)			0.36
Less than High School	(0, 0%)	(0, 0%)	--
High School or Equivalent	(5, 46%)	(3, 23%)	--
Associates’ Degree	(1, 9%)	(0, 0%)	
Bachelor’s Degree	(1, 9%)	(4, 31%)	--
Master’s Degree	(3, 27%)	(5, 38%)	--
Doctoral Degree	(1, 9%)	(1, 8%)	--
Household Income (*n*, %)			0.21
Less than USD 5000	(0, 0%)	(0, 0%)	--
USD 5000 to USD 11,999	(0, 0%)	(0, 0%)	--
USD 12,000–USD 15,999	(0, 0%)	(1, 8%)	--
USD 16,000–USD 24,999	(3, 28%)	(1, 8%)	--
USD 25,000–USD 34,999	(2, 18%)	(0, 0%)	--
USD 35,000–USD 49,999	(2, 18%)	(2, 15%)	--
USD 50,000–USD 74,999	(1, 9%)	(3, 23%)	--
USD 75,000 to USD 99,999	(1, 9%)	(1, 8%)	--
USD 100,000 and greater	(1, 9%)	(5, 38%)	--
Not reported	(1, 9%)	--	--
Currently employed (*n*, %)	(2, 18%)	(1, 8%)	0.88
Dosages taken (%)	97.07% (2.79%)	97.17% (2.30%)	0.46
CDR ^a^	0.5 (0.00)	0.5 (0.00)	--

*Note*. Sex refers to biological sex (M = male; F = female). Independent *t*-tests and chi-square analyses were used to determine significant differences between elderberry and placebo–control groups for continuous and categorical variables, respectively. CDR = Clinical Dementia Rating Scale; ^a^ cannot conduct group comparisons due to a constant value in both groups.

**Table 2 ijms-25-04352-t002:** Descriptive Values and Multilevel Modeling Results for Clinical and Cognitive Outcomes Across Time for American Elderberry Juice Intervention (*n* = 11) vs. Placebo–Control (*n* = 13).

	Baseline		3 Months		6 Months		Time		Condition		Time x Condition		
*Variable*	*M*	*SD*	*M*	*SD*	*M*	*SD*	*F*	*p*	*F*	*p*	*F*	*p*	*η* ^2^
MMSE							0.72	0.40	0.54	0.63	0.84	0.37	0.02
Elderberry	25.27	3.29	25.00	2.65	24.00	4.61	--	--	--	--	--	--	--
Placebo	24.62	3.31	25.27	3.93	25.00	2.79	--	--	--	--	--	--	--
HVLT—Free Recall							0.75	0.39	0.05	0.83	0.00	0.99	0.01
Elderberry	4.48	1.25	3.74	1.27	4.00	1.01	--	--	--	--	--	--	--
Placebo	4.04	1.63	4.54	2.09	3.83	1.77	--	--	--	--	--	--	--
HVLT—Delayed Free Recall							0.15	0.70	0.50	0.48	0.41	0.52	0.01
Elderberry	2.18	2.60	2.56	2.83	1.67	2.29	--	--	--	--	--	--	--
Placebo	1.46	1.98	2.45	3.39	1.90	3.03	--	--	--	--	--	--	--
HVLT—Recognition # Hits							3.35	0.08 *	0.00	0.96	2.32	0.14	0.05
Elderberry	11.27	0.79	11.38	0.74	11.14	1.07	--	--	--	--	--	--	--
Placebo	10.42	1.62	10.73	1.10	9.25	1.91	--	--	--	--	--	--	--
HVLT—Recognition FA-Related							0.56	0.46	0.01	0.91	0.13	0.73	0.01
Elderberry	2.27	1.79	2.38	2.13	2.86	1.95	--	--	--	--	--	--	--
Placebo	1.92	2.27	2.55	2.02	1.88	2.10	--	--	--	--	--	--	--
HVLT—Recognition FA-Unrelated							2.08	0.16	0.46	0.50	0.39	0.54	0.01
Elderberry	0.64	0.81	1.25	1.28	1.29	1.38	--	--	--	--	--	--	--
Placebo	0.92	1.31	1.27	2.00	1.12	1.73	--	--	--	--	--	--	--
HVLT—Discrimination Index							3.69	0.06 *	0.16	0.99	0.02	0.89	0.01
Elderberry	8.36	2.58	7.75	3.06	7.00	2.08	--	--	--	--	--	--	--
Placebo	7.58	3.45	6.91	3.24	6.25	3.45	--	--	--	--	--	--	--
BNT—Total Correct							8.48	0.006 ***	0.84	0.36	0.53	0.47	0.01
Elderberry	56.45	3.14	57.56	2.30	57.11	3.55	--	--	--	--	--	--	--
Placebo	55.23	2.83	55.91	4.44	56.55	3.67	--	--	--	--	--	--	--
REY CFT—Copy Total							0.55	0.47	0.17	0.68	0.12	0.74	0.01
Elderberry	30.55	3.70	29.06	8.15	29.21	3.91	--	--	--	--	--	--	--
Placebo	27.21	8.84	28.17	6.90	24.81	10.13	--	--	--	--	--	--	--
REY CFT—Delayed Copy Total							0.31	0.58	0.46	0.50	0.58	0.45	0.02
Elderberry	4.32	7.43	5.12	4.63	3.93	5.56	--	--	--	--	--	--	--
Placebo	3.29	5.30	2.50	4.89	4.12	5.42	--	--	--	--	--	--	--
Anagrams—Total							0.66	0.42	1.60	0.21	1.72	0.20	0.04
Elderberry	9.91	6.53	8.88	6.81	9.67	6.10	--	--	--	--	--	--	--
Placebo	6.85	3.85	10.18	3.74	9.18	4.29	--	--	--	--	--	--	--
Anagrams—Latency Total							0.13	0.72	0.73	0.99	1.06	0.31	0.02
Elderberry	18.54	12.49	15.10	6.76	16.91	9.89	--	--	--	--	--	--	--
Placebo	16.52	14.11	14.43	6.53	21.44	13.87	--	--	--	--	--	--	--
VPS—Total Correct							0.84	0.37	0.33	0.99	0.60	0.44	0.02
Elderberry	2.73	2.20	3.22	2.22	3.25	2.19	--	--	--	--	--	--	--
Placebo	2.46	1.85	3.00	2.00	3.27	2.05	--	--	--	--	--	--	--
VPS—Mean Latency Correct							0.77	0.38	2.11	0.15	3.33	0.07 *	0.06
Elderberry	43.31	34.41	33.71	25.05	23.20	14.45	--	--	--	--	--	--	--
Placebo	31.73	42.74	35.79	31.26	41.42	34.78	--	--	--	--	--	--	--

*Note*. For MLM interaction terms, η^2^ was used to describe effect sizes [small (η^2^ = 0.01), medium (η^2^ = 0.06), and large (η^2^ = 0.14); [36,37]; MMSE = Mini-Mental State Examination; HVLT = Hopkins Verbal Learning Test; FA = False Alarm; BNT = Boston Naming Test; REY CFT = Rey–Osterrieth Complex Figure Test; VPS = Visuospatial Problem Solving, # = Number. MLM models were also conducted with sex as a covariate. The sex covariate was non-significant, and results remained similar across all outcomes. Thus, it was not considered further, and results are presented without including sex in MLM models. * *p* < 0.10 (trending); *** *p* < 0.01

**Table 3 ijms-25-04352-t003:** Descriptive and Multilevel Modeling Results for Blood-Based Inflammatory Marker Outcomes for American Elderberry Juice Intervention (*n* = 7) vs. Placebo–Control (*n* = 5).

	Baseline		3 Months		6 Months		Time		Condition		Time x Condition		
*Variable*	*M*	*SD*	*M*	*SD*	*M*	*SD*	*F*	*p*	*F*	*p*	*F*	*p*	*η* ^2^
LDHA							7.68	0.01 **	6.71	0.02 **	5.63	0.03 **	0.11
Elderberry	2.35	1.45	2.06	0.62	2.14	1.17	--	--	--	--	--	--	--
Placebo	4.78	2.34	2.70	1.08	2.07	0.89	--	--	--	--	--	--	--
LDHB							5.70	0.03 **	2.52	0.12	4.27	0.05 *	0.10
Elderberry	2.45	1.34	3.44	0.94	3.36	1.58	--	--	--	--	--	--	--
Placebo	4.12	1.41	3.50	1.63	1.88	1.03	--	--	--	--	--	--	--
A2M							2.58	0.12	3.55	0.07 *	3.59	0.07 *	0.11
Elderberry	3119.87	492.68	3242.41	408.57	3573.79	514.29	--	--	--	--	--	--	--
Placebo	3872.48	656.22	4080.46	795.04	3834.99	645.91	--	--	--	--	--	--	--
Vasorin							2.63	0.12	3.22	0.08 *	4.26	0.05 *	0.12
Elderberry	5.44	2.20	3.71	1.08	3.53	2.38	--	--	--	--	--	--	--
Placebo	3.66	0.82	3.45	1.17	3.89	.85	--	--	--	--	--	--	--
PEDF							0.00	0.99	3.67	0.07 *	3.58	0.07 *	0.11
Elderberry	48.19	7.77	54.11	4.83	52.64	8.61	--	--	--	--	--	--	--
Placebo	58.37	4.03	56.13	7.32	53.90	7.30	--	--	--	--	--	--	--
C4-A							17.07	<0.001 ***	0.72	0.40	0.72	0.40	0.02
Elderberry	646.44	75.07	712.56	58.83	724.76	63.32	--	--	--	--	--	--	--
Placebo	736.38	104.29	777.38	92.29	788.00	67.93	--	--	--	--	--	--	--
C4-B							14.74	<0.001 ***	0.64	0.43	0.58	0.45	0.02
Elderberry	650.23	80.51	719.45	71.09	727.34	68.83	--	--	--	--	--	--	--
Placebo	742.91	106.47	784.59	88.91	794.49	70.93	--	--	--	--	--	--	--
PCYOX1							1.41	0.25	3.58	0.07 *	3.02	0.096 *	0.08
Elderberry	22.15	4.50	20.26	3.24	16.15	7.88	--	--	--	--	--	--	--
Placebo	16.19	5.25	15.42	2.22	17.32	5.02	--	--	--	--	--	--	--
Factor D							0.11	0.75	3.59	0.99	5.35	0.03 **	0.13
Elderberry	6.80	2.35	5.23	2.28	4.34	2.95	--	--	--	--	--	--	--
Placebo	5.67	1.60	4.52	1.37	7.52	1.79	--	--	--	--	--	--	--

*Note*. Inflammatory marker levels indicate the relative concentration. For MLM interaction terms, η^2^ was used to describe effect sizes [small (η^2^ = 0.01), medium (η^2^ = 0.06), and large (η^2^ = 0.14); [36,37]. LDHA = L-Lactate Dehydrogenase A Chain; LDHB = L-Lactate Dehydrogenase B Chain; A2M = Alpha-2-Macroglobulin; PEDF = Pigment Epithelium-Derived Factor; Complement C4-A; Complement C4-B; PCYOXY1 = Prenylcysteine Oxidase 1; Factor D = Complement Factor D. MLM models were also conducted with sex as a covariate. The sex covariate was non-significant, and results remained similar across all outcomes. Thus, it was not considered further, and results are presented without including sex in MLM models. * *p* < 0.10 (trending); ** *p* < 0.05; *** *p* < 0.01.

## Data Availability

Data can be requested by contacting the PI and corresponding author of the study (PI: Beversdorf).

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
