# Peer review of "Feasibility and Preliminary Efficacy of American Elderberry Juice for Improving Cognition and Inflammation in Patients with Mild Cognitive Impairment"

_ijms, 2024, doi:10.3390/ijms25084352_

Round 1
Reviewer 1 Report
Comments and Suggestions for Authors
The paper is both relevant and well suited to the aim and scopes of the special issue it is intended for. There are, however, a few things that could potentially benefit it as it is written:
1.- The authors make a commendable effort to thoroughly report the demographic variables of the experimental subjects, that, if extended to the 22 excluded subjects, would be even more informative about the circumstances of the enrolled participants. Also, the white-only ethnicity of the included participants can limit the impact of the results as shown, and I'd suggest the authors, probably unnecessarily, to try and broad the number and diversity of participants in the future.
2.- Sex is included as variable, which is always good to see in clinical paper, and I'd like to get some clarifications about it: is "sex" defined here as self-assigned gender identity, assigned sex at birth, or other? Also, as the number of participants in the elderberry cohort who completed the 6-month follow went from 11, 8 male and 3 female, to just 9, it could be useful to know the sexes of the participants at each stage, as 8 male to 1 female or 6 male to 3 female would potentially have very different interpretations.
3.- Given the importance of the nutritional component for the research present in the paper, a more detailed nutritional report of the elderberry juice employed in the study. I've tried to find one in the manufacturer's website and I could not, beyond a short paragraph with obvious commercial purposes. A more comprehensive nutritional analysis would make sense, particularly since the last paragraph in section 3, lines 420 to 426, point to the potential relevant interactions between elderberry and other nutritive factors affecting MCI and neuroinflammation.
4.- Plotting statistical data in a way that is appealing and makes sense to the reader is always challenging, more so when there are different cohorts and many relevant experimental results to show. Tables 1 to 3 are very well executed and clearly make the point, but figures 2 and 3 could perhaps be redesigned to be more homogeneous and, specially in figure 3, easier to read by increasing font size and removing the repeated Condition and Time legends, thus keeping a similar size to its current iteration.
Author Response
The paper is both relevant and well suited to the aim and scopes of the special issue it is intended for. There are, however, a few things that could potentially benefit it as it is written:
1.- The authors make a commendable effort to thoroughly report the demographic variables of the experimental subjects, that, if extended to the 22 excluded subjects, would be even more informative about the circumstances of the enrolled participants. Also, the white-only ethnicity of the included participants can limit the impact of the results as shown, and I'd suggest the authors, probably unnecessarily, to try and broad the number and diversity of participants in the future.
Response: Unfortunately, we do not have the demographic details of the 22 excluded participants, but agree this is an important area for consideration and reporting in future trials. To address the lack of diversity in the present study sample, we have also added the limitations the following:
“Finally, given that 100% of the study participants were White/Caucasian, it is critical that future studies examine the effects of elderberry juice in more racially and ethnically diverse samples.”
2.- Sex is included as variable, which is always good to see in clinical paper, and I'd like to get some clarifications about it: is "sex" defined here as self-assigned gender identity, assigned sex at birth, or other? Also, as the number of participants in the elderberry cohort who completed the 6-month follow went from 11, 8 male and 3 female, to just 9, it could be useful to know the sexes of the participants at each stage, as 8 male to 1 female or 6 male to 3 female would potentially have very different interpretations.
Response: We have clarified the following in Table 1 note:
“Sex refers to biological sex (M=male; F=female).”
We also corrected the reported sex distribution for the elderberry condition at baseline to 3 men/8 women and now also state the sex distribution for each condition at 3 and 6-month follow-ups in the Feasibility results:
“All remaining participants completed 3-month (elderberry: 3 men, 6 women; placebo-control: 6 men, 5 women) and 6-month follow-ups.”
“For blood-based proteomic outcomes, out of the 12 participants (6 men, 6 women) in the sub-sample who were asked to complete blood draws across the 3 timepoints, completion was very high in the elderberry condition [out of 7 participants (3 men, 4 women) in this condition, average of 100% of blood-draws completed, with a range of 100-100% across participants) and placebo-control [out of 5 participants (3 men, 2 women)”
3.- Given the importance of the nutritional component for the research present in the paper, a more detailed nutritional report of the elderberry juice employed in the study. I've tried to find one in the manufacturer's website and I could not, beyond a short paragraph with obvious commercial purposes. A more comprehensive nutritional analysis would make sense, particularly since the last paragraph in section 3, lines 420 to 426, point to the potential relevant interactions between elderberry and other nutritive factors affecting MCI and neuroinflammation.
Response: We now provide a more comprehensive nutritional analysis of the American Elderberry juice used in the present study. We have added 2 tables to the supplemental material. Formal analysis of the juice was performed by our collaborators.
4.- Plotting statistical data in a way that is appealing and makes sense to the reader is always challenging, more so when there are different cohorts and many relevant experimental results to show. Tables 1 to 3 are very well executed and clearly make the point, but figures 2 and 3 could perhaps be redesigned to be more homogeneous and, specially in figure 3, easier to read by increasing font size and removing the repeated Condition and Time legends, thus keeping a similar size to its current iteration.
Response: These figures have been updated.
Reviewer 2 Report
Comments and Suggestions for Authors
Curtis and colleagues provide a manuscript entitled “Feasibility and preliminary efficacy of American elderberry juice for improving cognition and inflammation in patients with mild cognitive impairment “. Despite these findings would be of general interest to this field of research, some points need to be addressed.
Major points
- The rationale for conducting this study is poorly discussed. Authors must better discuss the positive impact of nutritional interventions high in antioxidant/anti-inflammatory properties (anthocyanin-rich foods, such as blueberries/elderberries) not only on cognitive functions but in general on mental health (DOI: 10.3233/JBR-220054; DOI: 10.1080/14786419.2023.2275275 and others).
- Many statements are unclear. For example in the abstract: Elderberry also showed significant or significantly trending decreases over…
- Did the Authors consider the factor gender in the statistical analysis? This is crucial because the factor sex/gender may influences cognitive functions, as suggested by clinical and preclinical findings.
- The first part of the discussion is a reiteration of the results. The Authors must frankly discuss their findings according to the available literature.
Minor points
- There are several typos throughout the manuscript. For example in the abstract line 37: deceased
- Please check for statements without references throughout the manuscript.
Comments on the Quality of English LanguageModerate editing
Author Response
Curtis and colleagues provide a manuscript entitled “Feasibility and preliminary efficacy of American elderberry juice for improving cognition and inflammation in patients with mild cognitive impairment “. Despite these findings would be of general interest to this field of research, some points need to be addressed.
Major points
- The rationale for conducting this study is poorly discussed. Authors must better discuss the positive impact of nutritional interventions high in antioxidant/anti-inflammatory properties (anthocyanin-rich foods, such as blueberries/elderberries) not only on cognitive functions but in general on mental health (DOI: 10.3233/JBR-220054; DOI: 10.1080/14786419.2023.2275275 and others).
Response: Thank you for this comment. We aimed to focus the introduction on the relevant literature on the effects of anthocyanin rich nutritional interventions on our outcomes of interest (cognition, inflammation). The authors raise an important an important point that studies examining mental health may also be important to mention. We have added to the Introduction Paragraph 1:
“Elderberries have anti-inflammatory and antioxidant properties [7, 8], and show promising potential effects for improving cognition [9, 10] and other areas of mental health [1-3] and related functioning [1, 4].”
We have also added to the limitations section the following areas for future research:
“On a related note, given the known impact of nutritional interventions rich in anthocyanin/anti-inflammatory properties on other aspects of mental health (depression, anxiety and stress [1-3]) and sleep health [1, 4] in adults without cognitive impairment, future studies in MCI should also examine the impact of American elderberry juice on a broader range of outcomes beyond cognition and inflammation.”
- Many statements are unclear. For example in the abstract: Elderberry also showed significant or significantly trending decreases over…
Response: We have clarified this sentence:
“For the Elderberry condition, there were significant or significantly trending decreases over time across several markers of low-grade peripheral inflammation, including vasorin, prenylcysteine oxidase 1 and complement Factor D. Only one inflammatory marker showed an increase over time (alpha-2-macroglobin).”
- Did the Authors consider the factor gender in the statistical analysis? This is crucial because the factor sex/gender may influences cognitive functions, as suggested by clinical and preclinical findings.
Response: We re-ran analyses with sex as a covariate in MLM and it did not alter the pattern of results. Thus, we have not included this in results reporting. We have however added to the note on Table 2 and 3 the following:
“MLM models were also conducted with sex as a covariate. The sex covariate was non-significant, and results remained similar across all outcomes. Thus, it was not considered further, and results are presented without including sex in MLM models.”
We now also state in the limitations/future directions the following:
“Additionally, given known sex differences in trajectories of cognition [5, 6] and low-grade inflammation [7] in older adults, future work should also examine sex-specific impact of elderberry juice on cognitive and inflammation outcomes.”
- The first part of the discussion is a reiteration of the results. The Authors must frankly discuss their findings according to the available literature.
Response: Given the large range of outcomes, we felt it necessary to include an overall summary paragraph in the beginning of the discussion. We then go on to discuss whether hypotheses are supported and discuss findings relative to available literature in the remaining paragraphs in the discussion.
Minor points
- There are several typos throughout the manuscript. For example in the abstract line 37: deceased
Response: This has been corrected and the manuscript has been proofed to correct other typos.
- Please check for statements without references throughout the manuscript.
Response: These have been checked.
Round 2
Reviewer 2 Report
Comments and Suggestions for Authors
The Authors have addressed most of the points I raised.
Comments on the Quality of English Languageminor editing